# From MAP to Marginals: Variational Inference in Bayesian Submodular Models

**Josip Djolonga**
Department of Computer Science
ETH Zürich
josipd@inf.ethz.ch

**Andreas Krause**
Department of Computer Science
ETH Zürich
krausea@ethz.ch

## Abstract

Submodular optimization has found many applications in machine learning and beyond. We carry out the first systematic investigation of inference in probabilistic models defined through submodular functions, generalizing regular pairwise MRFs and Determinantal Point Processes. In particular, we present L-FIELD, a variational approach to general log-submodular and log-supermodular distributions based on sub- and supergradients. We obtain *both* lower and upper bounds on the log-partition function, which enables us to compute probability intervals for marginals, conditionals and marginal likelihoods. We also obtain fully factorized approximate posteriors, at the same computational cost as ordinary submodular optimization. Our framework results in convex problems for optimizing over differentials of submodular functions, which we show how to optimally solve. We provide theoretical guarantees of the approximation quality with respect to the curvature of the function. We further establish natural relations between our variational approach and the classical mean-field method. Lastly, we empirically demonstrate the accuracy of our inference scheme on several submodular models.

## 1 Introduction

Submodular functions [1] are a rich class of set functions $F : 2^V \to \mathbb{R}$, investigated originally in game theory and combinatorial optimization. They capture natural notions such as diminishing returns and economies of scale. In recent years, submodular optimization has seen many important applications in machine learning, including active learning [2], recommender systems [3], document summarization [4], representation learning [5], clustering [6], the design of structured norms [7] etc. In this work, instead of using submodular functions to obtain point estimates through optimization, we take a Bayesian approach and define probabilistic models over sets (so called point processes) using submodular functions. Many of the aforementioned applications can be understood as performing MAP inference in such models. We develop L-FIELD, a general variational inference scheme for reasoning about *log-supermodular* ($P(A) \propto \exp(-F(A))$) and *log-submodular* ($P(A) \propto \exp(F(A))$) distributions, where $F$ is a submodular set function.

**Previous work.** There has been extensive work on submodular optimization (both approximate and exact minimization and maximization, see, e.g., [8, 9, 10, 11]). In contrast, we are unaware of previous work that addresses the general problem of probabilistic inference in Bayesian submodular models. There are two important special cases that have received significant interest. The most prominent examples are undirected pairwise Markov Random Fields (MRFs) with binary variables, also called the Ising model [12], due to their importance in statistical physics, and applications, e.g., in computer vision. While MAP inference is efficient for regular (log-supermodular) MRFs, computing the partition function is known to be #P-hard [13], and the approximation problem has been also shown to be hard [14]. Also, there is no FPRAS in the log-submodular case unless RP=NP [13]. An important case of log-submodular distributions is the *Determinantal Point Process (*DPP*)*, used

in machine learning as a principled way of modeling diversity. Its partition function can be computed efficiently, and a $\frac{1}{4}$-approximation scheme for finding the (NP-hard) MAP [15] is known. In this paper, we propose a variational inference scheme for general Bayesian submodular models, that encompasses these two and many other distributions, and has instance-dependent quality guarantees. A hallmark of the models is that they capture high-order interactions between many random variables. Existing variational approaches [16] cannot efficiently cope with such high-order interactions — they generally have to sum over all variables in a factor, scaling exponentially in the size of the factor. We discuss this prototypically for mean-field in Sec. 5.

**Our contributions.** In summary, our main contributions are

- We provide the first general treatment of probabilistic inference with log-submodular and log-supermodular distributions, that can capture *high-order variable interactions*.
- We develop L-FIELD, a novel variational inference scheme that optimizes over sub- and supergradients of submodular functions. Our scheme yields both upper and lower bounds on the partition function, which imply rigorous probability intervals for marginals. We can also obtain factorial approximations of the distribution at no larger computational cost than performing MAP inference in the model (for which a plethora of algorithms are available).
- We identify a natural link between our scheme and the well-known mean-field method.
- We establish theoretical guarantees about the accuracy of our bounds, dependent on the curvature of the underlying submodular function.
- We demonstrate the accuracy of L-FIELD on several Bayesian submodular models.

## 2 Submodular functions and optimization

Submodular functions are set functions satisfying a diminishing returns condition. Formally, let $V$ be some finite ground set, w.l.o.g. $V = \{1, \ldots, n\}$, and consider a set function $F : 2^V \to \mathbb{R}$. The marginal gain of adding item $i \in V$ to the set $A \subseteq V$ w.r.t. $F$ is defined as $F(i|A) = F(A \cup \{i\}) - F(A)$. Then, a function $F : 2^V \to \mathbb{R}$ is said to be *submodular* if for all $A \subseteq B \subseteq V$ and $i \in V - B$ it holds that $F(i|A) \geq F(i|B)$. A function $F$ is called *supermodular* if $-F$ is submodular. Without loss of generality[1], we will also make the assumption that $F$ is *normalized* so that $F(\emptyset) = 0$.

The problem of submodular function optimization has received significant attention. The (unconstrained) *minimization* of submodular functions, $\min_A F(A)$, can be done in polynomial time. While general purpose algorithms [8] can be impractical due to their high order, several classes of functions admit faster, specialized algorithms, e.g. [17, 18, 19]. Many important problems can be cast as the minimization of a submodular objective, ranging from image segmentation [20, 12] to clustering [6]. Submodular *maximization* has also found numerous applications, e.g. experimental design [21], document summarization [4] or representation learning [5]. While this problem is in general NP-hard, effective constant-factor approximation algorithms exist (e.g. [22, 11]).

In this paper we lift results from submodular optimization to probabilistic inference, which lets us quantify uncertainty about the solutions of the problem, instead of binding us to a single one. Our approach allows us to obtain (approximate) marginals at the same cost as traditional MAP inference.

## 3 Probabilistic inference in Bayesian submodular models

Which Bayesian models are associated with submodular functions? Suppose $F : 2^V \to \mathbb{R}$ is a submodular set function. We consider distributions over subsets[2] $A \subseteq V$ of the form $P(A) = \frac{1}{\mathcal{Z}} e^{+F(A)}$ and $P(A) = \frac{1}{\mathcal{Z}} e^{-F(A)}$, which we call log-submodular and log-supermodular, respectively. The normalizing quantity $\mathcal{Z} = \sum_{S \subseteq V} e^{\pm F(S)}$ is called the *partition function*, and $-\log \mathcal{Z}$ is also known as *free energy* in the statistical physics literature. Note that distributions over subsets of $V$ are isomorphic to distributions of $|V| = n$ binary random variables $X_1, \ldots, X_n \in \{0, 1\}$ — we simply identify $X_i$ as the indicator function of the event $i \in A$, or formally $X_i = [i \in A]$.

**Examples of log-supermodular distributions.** There are many distributions that fit this framework. As a prominent example, consider *binary pairwise Markov random fields (MRFs)*,

$P(X_1, \ldots, X_n) = \frac{1}{\mathcal{Z}} \prod_{i,j} \phi_{i,j}(X_i, X_j)$. Assuming the potentials $\phi_{i,j}$ are positive, such MRFs are equivalent to distributions $P(A) \propto \exp(-F(A))$, where $F(A) = \sum_{i,j} F_{i,j}(A)$, and $F_{i,j}(A) = -\log \phi_{i,j}([i \in A], [j \in A])$. An MRF is called *regular* iff each $F_{i,j}$ is submodular (and consequently $P(A)$ is log-supermodular). Such models are extensively used in applications, e.g. in computer vision [12]. More generally, a rich class of distributions can be defined using *decomposable submodular functions*, which can be written as sums of (usually simpler) submodular functions. As an example, let $G_1, \ldots, G_k \subseteq V$ be groups of elements and let $\phi_1, \ldots, \phi_k : [0, \infty) \to \mathbb{R}$ be concave. Then, the function $F(A) = \sum_{i=1}^{k} \phi_i(|G_i \cap A|)$ is submodular. Models using these types of functions strictly generalize pairwise MRFs, and can capture higher-order variable interactions, which can be crucial in computer vision applications such as semantic segmentation (e.g. [23]).

**Examples of log-submodular distributions.** A prominent example of log-submodular distributions are *Determinantal Point Processes (*DPP*s)* [24]. A DPP is a distribution over sets $A$ of the form $P(A) = \frac{1}{\mathcal{Z}} \exp(F(A))$, where $F(A) = \log |K_A|$. Here, $K \in \mathbb{R}^{V \times V}$ is a positive semi-definite matrix, $K_A$ is the square submatrix indexed by $A$, and $|\cdot|$ denotes the determinant. Because $K$ is positive semi-definite, $F(A)$ is known to be submodular, and hence DPPs are log-submodular. Another natural model is that of facility location. Assume that we have a set of locations $V$ where we can open shops, and a set $N$ of customers that we would like to serve. For each customer $i \in N$ and location $j \in V$ we have a non-negative number $C_{i,j}$ quantifying how much service $i$ gets from location $j$. Then, we consider $F(A) = \sum_{i \in N} \max_{j \in A} C_{i,j}$. We can also penalize the number of open shops and use a distribution $P(A) \propto \exp(F(A) - \lambda|A|)$ for $\lambda > 0$. Such objectives have been used for optimization in many applications, ranging from clustering [25] to recommender systems [26].

**The Inference Challenge.** Having introduced the models that we consider, we now show how to do inference in them[3]. Let us introduce the following operations that preserve submodularity.

**Definition 1.** *Let* $F : 2^V \to \mathbb{R}$ *be submodular and let* $X, Y \subseteq V$. *Define the submodular functions* $F^X$ *as the restriction of* $F$ *to* $2^X$, *and* $F_X : 2^{V-X} \to \mathbb{R}$ *as* $F_X(A) = F(A \cup X) - F(X)$.

First, let us see how to compute marginals. The probability that the random subset $\mathbf{S}$ distributed as $P(\mathbf{S} = A) \propto \exp(-F(A))$ is in some non-empty lattice $[X, Y] = \{A \mid X \subseteq A \subseteq Y\}$ is equal to

$$P(\mathbf{S} \in [X, Y]) = \frac{1}{\mathcal{Z}} \sum_{X \subseteq A \subseteq Y} \exp(-F(A)) = \frac{1}{\mathcal{Z}} \sum_{A \subseteq Y - X} \exp(-F(X \cup A)) = e^{-F(X)} \frac{\mathcal{Z}_X^Y}{\mathcal{Z}}, \quad (1)$$

where $\mathcal{Z}_X^Y = \sum_{A \subseteq Y - X} e^{-(F(X \cup A) - F(X))}$ is the partition function of $(F_X)^Y$. Marginals $P(i \in \mathbf{S})$ of any $i \in V$ can be obtained using $[\{i\}, V]$. We also obtain conditionals — if, for example, we condition on the event on (1), we have $P(\mathbf{S} = A | \mathbf{S} \in [X, Y]) = \exp(-F(A))/\mathcal{Z}_X^Y$ if $A \in [X, Y]$, 0 otherwise. Note that log-supermodular distributions are conjugate with each other: for a log-supermodular prior $P(A) \propto \exp(-F(A))$ and a likelihood function[4] $P(E \mid A) \propto \exp(-L(E; A))$, for which $L$ is submodular w.r.t. $A$ for each evidence $E$, the posterior $P(A \mid E) \propto \exp(-(F(A) + L(E; A)))$ is log-supermodular as well. The same holds for log-submodular distributions.

# 4 The variational approach

In Section 3 we have seen that due to the closure properties of submodular functions, important inference tasks (e.g., marginals, conditioning) in Bayesian submodular models require computing partition functions of suitably defined/restricted submodular functions. Given that the general problem is #P hard, we seek approximate methods. The main idea is to exploit the peculiar property of submodular functions that they can be *both lower- and upper-bounded* using simple additive functions of the form $s(A) + c$, where $c \in \mathbb{R}$ and $s : 2^V \to \mathbb{R}$ is *modular*, i.e. it satisfies $s(A) = \sum_{i \in A} s(\{i\})$. We will also treat modular functions $s(\cdot)$ as vectors $\mathbf{s} \in \mathbb{R}^V$ with coordinates $s_i = s(\{i\})$. Because modular functions have tractable log-partition functions, we obtain the following bounds.

**Lemma 1.** *If* $\forall A \subseteq V : s_l(A) + c_l \leq F(A) \leq s_u(A) + c_u$ *for modular* $\mathbf{s}_u, \mathbf{s}_l$, *and* $c_l, c_u \in \mathbb{R}$, *then*

$$\begin{aligned} \log \mathcal{Z}^+(\mathbf{s}_l, c_l) &\leq \log \sum_{A \subseteq V} \exp(+F(A)) \leq \log \mathcal{Z}^+(\mathbf{s}_u, c_u) \text{ and} \\ \log \mathcal{Z}^-(\mathbf{s}_u, c_u) &\leq \log \sum_{A \subseteq V} \exp(-F(A)) \leq \log \mathcal{Z}^-(\mathbf{s}_l, c_l), \end{aligned}$$

*where* $\log \mathcal{Z}^+(\mathbf{s}, c) = c + \sum_{i \in V} \log(1 + e^{s_i})$ *and* $\log \mathcal{Z}^-(\mathbf{s}, c) = -c + \sum_{i \in V} \log(1 + e^{-s_i})$.

We can use any modular (upper or lower) bound $s(A) + c$ to define a completely factorized distribution that can be used as a proxy to approximate values of interest of the original distribution. For example, the marginal of $i \in A$ under $Q(A) \propto \exp(-s(A) + c)$ is easily seen to be $1/(1 + e^{s_i})$.

Instead of optimizing over all possible bounds of the above form, we consider for each $X \subseteq V$ two sets of modular functions, which are exact at $X$ and lower- or upper-bound $F$ respectively. Similarly as for convex functions, we define [8][§6.2] the *subdifferential* of $F$ at $X$ as

$$\partial_F(X) = \{\mathbf{s} \in \mathbb{R}^n \mid \forall Y \subseteq V \colon F(Y) \geq F(X) + \mathbf{s}(Y) - \mathbf{s}(X)\}. \tag{2}$$

The *superdifferential* $\partial^F(X)$ is defined analogously by inverting the inequality sign [27]. For each *subgradient* $\mathbf{s} \in \partial_F(X)$, the function $g_X(Y) = s(Y) + F(X) - s(X)$ is *lower* bounding $F$. Similarly, for a *supergradient* $\mathbf{s} \in \partial^F(X)$, $h_X(Y) = s(Y) + F(X) - s(X)$ is an *upper* bound of $F$. Note that both $h_X$ and $g_X$ are of the form that we considered (modular plus constant) and are tight at $X$, i.e. $h_X(X) = g_X(X) = F(X)$. Because we will be optimizing over differentials, we define for any $X \subseteq V$ the shorthands $\mathcal{Z}_X^+(\mathbf{s}) = \mathcal{Z}^+(\mathbf{s}, F(X) - s(X))$ and $\mathcal{Z}_X^-(\mathbf{s}) = \mathcal{Z}^-(\mathbf{s}, F(X) - s(X))$.

### 4.1 Optimizing over subgradients

To analyze the problem of minimizing $\log \mathcal{Z}_X^-(\mathbf{s})$ subject to $\mathbf{s} \in \partial_F(X)$, we introduce the *base polyhedron* of $F$, defined as $B(F) = \{\mathbf{s} \in \mathbb{R}^V \mid s(V) = F(V) \text{ and } \forall A \subseteq V \colon s(A) \leq F(A)\}$, i.e. the set of modular lower bounds that are exact at $V$. As the following lemma shows, we do not have to consider $\log \mathcal{Z}_X^-$ for all $X$ and we can restrict our attention to the case $X = \emptyset$.

**Lemma 2.** *For all $X \subseteq V$ we have $\min_{\mathbf{s} \in \partial_F(\emptyset)} \mathcal{Z}_\emptyset^-(\mathbf{s}) \leq \min_{\mathbf{s} \in \partial_F(X)} \mathcal{Z}_X^-(\mathbf{s})$. Moreover, the former problem is equivalent to*

$$\underset{\mathbf{s}}{\text{minimize}} \sum_{i \in V} \log(1 + e^{-s_i}) \quad \text{subject to} \quad \mathbf{s} \in B(F). \tag{3}$$

Thus, we have to optimize a convex function over $B(F)$, a problem that has been already considered [8, 9]. For example, we can use the Frank-Wolfe algorithm [28, 29], which is easy to implement and has a convergence rate of $O(\frac{1}{k})$. It requires the optimization of linear functions $g(\mathbf{s}) = \langle \mathbf{w}, \mathbf{s} \rangle = \mathbf{w}^T \mathbf{s}$ over the domain, which, as shown by Edmonds [1], can be done greedily in $O(|V| \log |V|)$ time. More precisely, to compute a maximizer $\mathbf{s}^* \in B(F)$ of $g(\mathbf{s})$, pick a bijection $\sigma \colon \{1, \ldots, |V|\} \to V$ that orders $\mathbf{w}$, i.e. $w_{\sigma(1)} \geq w_{\sigma(2)} \geq \cdots \geq w_{\sigma(|V|)}$. Then, set $s_{\sigma(i)}^* = F(\sigma(i)|\{\sigma(1), \ldots, \sigma(i-1)\})$. Alternatively, if we can efficiently minimize the sum of the function plus a modular term, e.g. for the family of graph-cut representable functions [10], we can apply the divide-and-conquer algorithm [9][§9.1], which needs the minimization of $O(|V|)$ problems.

```
 1: procedure FRANK-WOLFE(F, x¹, ε)
 2:     Define f(x) = log(1 + e⁻ˣ)    ▷ Elementwise.
 3:     for k ← 1, 2, ..., T do
 4:         Pick s ∈ argmin_{x∈B(F)}⟨x, ∇f(xᵏ)⟩
 5:         if ⟨xᵏ - s, ∇f(xᵏ)⟩ ≤ ε then
 6:             return xᵏ              ▷ Small duality gap.
 7:         else
 8:             x^{k+1} = (1 - γₖ)xᵏ + γₖs;  γₖ = 2/(k+2)
```

```
 1: procedure DIVIDE-CONQUER(F)
 2:     s ← (F(V)/|V|)1;  A* ← minimizer of F(·) - s(·)
 3:     if F(A*) = s(A*) then
 4:         return s
 5:     else
 6:         s_A ←DIVIDE-CONQUER(F^A)
 7:         s_{V-A} ←DIVIDE-CONQUER(F_A)
 8:         return (s_A, s_{V-A})
```

**The entropy viewpoint and the Fenchel dual.** Interestingly, (3) can be interpreted as a maximum entropy problem. Recall that, for $\mathbf{s} \in B(F)$ we use the distribution $P(A) \propto \exp(-s(A))$, whose entropy is exactly the negative of our objective. Hence, we can consider Problem (3) as that of maximizing the entropy over the set of factorized distributions with parameters in $-B(F)$. We can go back to the standard representation using the marginals $\mathbf{p}$ via $p_i = 1/(1 + \exp(s_i))$. This becomes obvious if we consider the Fenchel dual of the problem, which, as discussed in §5, allows us to make connections with the classical mean-field approach. To this end, we introduce the Lovàsz extension, defined for any $F \colon 2^V \to \mathbb{R}$ as the support function over $B(F)$, i.e. $f(\mathbf{p}) = \sup_{s \in B(F)} \mathbf{s}^T \mathbf{p}$ [30].

Let us also define for $\mathbf{p} \in [0, 1]^V$ by $\mathbb{H}[\mathbf{p}]$ the Shannon entropy of a vector of $|V|$ independent Bernoulli random variables with success probabilities $\mathbf{p}$.

**Lemma 3.** *The Fenchel dual problem of Problem* (3) *is*

$$\underset{\mathbf{p}\in[0,1]^V}{\text{maximize}} \quad \mathbb{H}[\mathbf{p}] - f(\mathbf{p}). \tag{4}$$

*Moreover, there is zero duality gap, and the pair* $(\mathbf{s}^*, \mathbf{p}^*)$ *is primal-dual optimal if and only if*

$$\mathbf{p}^* = \Big(\frac{1}{1+\exp(s_i^*)}, \dots, \frac{1}{1+\exp(s_n^*)}\Big) \quad and \quad f(\mathbf{p}^*) = \mathbf{p}^{*T}\mathbf{s}^*. \tag{5}$$

From the discussion above, it can be easily seen that the Fenchel dual reparameterizes the problem from the parameters $-\mathbf{s}$ to the marginals $\mathbf{p}$. Note that the dual lets us provide a certificate of optimality, as the Lovász extension can be computed with Edmonds' greedy algorithm.

## 4.2 Optimizing over supergradients

To optimize over subgradients, we pick for each set $X \subseteq V$ a representative supergradient and optimize over all $X$. As in [27], we consider the following supergradients, elements of $\partial^F(X)$.

| | Grow supergradient $\hat{\mathbf{s}}^X$ | Shrink supergradient $\check{\mathbf{s}}^X$ | Bar supergradient $\bar{\mathbf{s}}^X$ |
|---|---|---|---|
| $i \in X$ | $\hat{\mathbf{s}}^X(\{i\}) = F(i\|V - \{i\})$ | $\check{\mathbf{s}}^X(\{i\}) = F(i\|X - \{i\})$ | $\bar{\mathbf{s}}^X(\{i\}) = F(i\|V - \{i\})$ |
| $i \notin X$ | $\hat{\mathbf{s}}^X(\{i\}) = F(i\|X)$ | $\check{\mathbf{s}}^X(\{i\}) = F(\{i\})$ | $\bar{\mathbf{s}}^X(\{i\}) = F(\{i\})$ |

Optimizing the bound over *bar* supergradients requires the minimization of the original function plus a modular term. As already mentioned for the divide-and-conquer strategy above, we can do this efficiently for several problems. The exact formulation of the problem is presented below.

**Lemma 4.** *Define the modular functions* $m_1(\{i\}) = \log(1 + e^{-F(i|V-i)}) - \log(1 + e^{F(i)})$, *and* $m_2(\{i\}) = \log(1 + e^{F(i|V-i)}) - \log(1 + e^{-F(i)})$. *The following pairs of problems are equivalent.*

$$\begin{aligned}
\text{minimize}_X \log \mathcal{Z}_X^+(\bar{\mathbf{s}}^X) &\equiv \text{minimize}_X F(X) + m_1(X) \\
\text{maximize}_X \log \mathcal{Z}_X^-(\bar{\mathbf{s}}^X) &\equiv \text{minimize}_X F(X) - m_2(X)
\end{aligned}$$

Even though we cannot optimize over *grow* and *shrink* supergradients, we can evaluate all three at the optimum for the problems above and pick the one that gives the best bound.

## 5 Mean-field methods and the multi-linear extension

Is there a relation to traditional variational methods? If $Q(\cdot)$ is a distribution over subsets of $V$, then

$$0 \le \mathbb{KL}(Q \,\|\, P) = \mathbb{E}_Q\Big[\log \frac{Q(\mathbf{S})}{P(\mathbf{S})}\Big] = \log \mathcal{Z} + \mathbb{E}_Q\Big[\log \frac{Q(\mathbf{S})}{\exp(-F(\mathbf{S}))}\Big] = \log \mathcal{Z} - \mathbb{H}[Q] + \mathbb{E}_Q[F],$$

which yields the bound $\log \mathcal{Z} \ge \mathbb{H}[Q] - \mathbb{E}_Q[F]$. The *mean-field* method restricts $Q$ to be a completely factorized distribution, so that elements are picked independently and $Q$ can be described by the vector of marginals $\mathbf{q} \in [0,1]^V$, over which it is then optimized. Compare this with our approach.

| Mean-Field Objective | Our Objective: L-FIELD |
|---|---|
| maximize$_{\mathbf{q}\in[0,1]^V}$ $\mathbb{H}[\mathbf{q}] - \mathbb{E}_{\mathbf{q}}[F]$ | maximize$_{\mathbf{q}\in[0,1]^V}$ $\mathbb{H}[\mathbf{q}] - f(\mathbf{q})$ |
| ▷ *Non-concave, can be hard to evaluate.* | ▷ *Concave, efficient to evaluate.* |

Both the Lovász extension $f(\mathbf{q})$ and the multi-linear extension $\tilde{f}(\mathbf{q}) = \mathbb{E}_{\mathbf{q}}[F]$ are continuous extensions of $F$, introduced for submodular minimization [30] and maximization [31], respectively. The former agrees with the convex envelope of $F$ and can be efficiently evaluated (in $O(|V|)$ evaluations of $F$) using Edmonds' greedy algorithm (cf., §4.1, [1]). In contrast, evaluating $\tilde{f}(\mathbf{q}) = \mathbb{E}_{\mathbf{q}}[F] = \sum_{A \subseteq V} \prod_i q_i^{[i\in A]}(1-q_i)^{[i\notin A]}F(A)$ in general requires summing over exponentially many terms – a problem potentially as hard as the original inference problem! Even if $\tilde{f}(\mathbf{q})$ is approximated by sampling, it is neither convex nor concave. Moreover, computing the coordinate ascent updates of mean-field can be intractable for general $F$. Hence, our approach can be motivated as follows: instead of using the multi-linear extension $\tilde{f}$, we use the Lovász extension $f$ of $F$, which makes the problem convex and tractable. This analogy motivated the name L-FIELD (L for Lovász).

# 6  Curvature-dependent approximation bounds

How accurate are the bounds obtained via our variational approach? We now provide theoretical guarantees on the approximation quality as a function of the *curvature* of $F$, which quantifies how far the function is from modularity. Curvature is defined for *polymatroid* functions, which are normalized non-decreasing submodular functions, i.e., a submodular function $F : 2^V \to \mathbb{R}$ is polymatroid if for all $A \subseteq B \subseteq V$ it holds that $F(A) \leq F(B)$.

**Definition 2** (From [32]). *Let $G : 2^V \to \mathbb{R}$ be a polymatroid function. The curvature $\kappa$ of $G$ is defined as* [5] $\kappa = 1 - \min_{i \in V \,:\, G(\{i\}) > 0} \frac{G(i|V - \{i\})}{G(\{i\})}$.

The curvature is always between 0 and 1 and is equal to 0 if and only if the function is modular. Although the curvature is a notion for polymatroid functions, we can still show results for the general case as any submodular function $F$ can be decomposed [33] as the sum of a modular term $m(\cdot)$ defined as $m(\{i\}) = F(i|V - \{i\})$ and $G = F - m$, which is a polymatroid function. Our bounds below depend on the curvature of $G$ and $G_{\text{MAX}} = G(V) = F(V) - \sum_{i \in V} F(i|V - i)$.

**Theorem 1.** *Let $F = G + m$, where $G$ is polymatroid with curvature $\kappa$ and $m$ is modular defined as above. Pick any bijection $\sigma : V \to \{1, 2, \ldots, |V|\}$ and define sets $S_0^\sigma = \emptyset, S_i^\sigma = \{\sigma(1), \ldots, \sigma(i)\}$. If we define $\mathbf{s} \colon s_{\sigma(i)} = G(S_i^\sigma) - G(S_{i-1}^\sigma)$, then $\mathbf{s} + \mathbf{m} \in \partial_F(\emptyset)$ and the following inequalities hold.*

$$\log \mathcal{Z}^-(\mathbf{s} + \mathbf{m}, 0) - \log \sum_{A \subseteq V} \exp(-F(A)) \leq \kappa G_{\text{MAX}} \tag{6}$$

$$\log \sum_{A \subseteq V} \exp(+F(A)) - \log \mathcal{Z}^+(\mathbf{s} + \mathbf{m}, 0) \leq \kappa G_{\text{MAX}} \tag{7}$$

**Theorem 2.** *Under the same assumptions as in Theorem 1, if we define the modular function $s(\cdot)$ by $s(A) = \sum_{i \in A} G(\{i\})$, then $\mathbf{s} + \mathbf{m} \in \partial^F(\emptyset)$ and the following inequalities hold.*

$$\log \sum_{A \subseteq V} \exp(-F(A)) - \log \mathcal{Z}^-(\mathbf{s} + \mathbf{m}, 0) \leq \frac{\kappa(n-1)}{1 + (n-1)(1-\kappa)} G_{\text{MAX}} \leq \frac{\kappa}{1-\kappa} G_{\text{MAX}} \tag{8}$$

$$\log \mathcal{Z}^+(\mathbf{s} + \mathbf{m}, 0) - \log \sum_{A \subseteq V} \exp(+F(A)) \leq \frac{\kappa(n-1)}{1 + (n-1)(1-\kappa)} G_{\text{MAX}} \leq \frac{\kappa}{1-\kappa} G_{\text{MAX}} \tag{9}$$

Note that we establish bounds for *specific* sub-/supergradients. Since our variational scheme considers these in the optimization as well, the same quality guarantees hold for the optimized bounds. Further, note that we get a dependence on the range of the function via $G_{\text{MAX}}$. However, if we consider $\alpha F$ for large $\alpha > 1$, most of the mass will be concentrated at the MAP (assuming it is unique). In this case, L-FIELD also performs well, as it can always choose gradients that are tight at the MAP. When we optimize over supergradients, all possible tight sets are considered. Similarly, the subgradients are optimized over $B(F)$, and for any $X \subseteq V$ there exists some $\mathbf{s}_X \in B(F)$ tight at $X$.

# 7  Experiments

Our experiments[6] aim to address four main questions: (1) How large is the gap between the upper- and lower-bounds for the log-partition function and the marginals? (2) How accurate are the factorized approximations obtained from a *single* MAP-like optimization problem? (3) How does the accuracy depend on the amount of evidence (i.e., concentration of the posterior), the curvature of the function, and the type of Bayesian submodular model considered? (4) How does L-FIELD compare to *mean-field* on problems where the latter can be applied?

We consider approximate marginals obtained from the following methods: *lower/upper*: obtained from the factorized distributions associated with the modular lower/upper bounds; *lower-/upper-bound*: the lower/upper bound of the estimated probability interval. All of the functions we consider are graph-representable [17], which allows us to perform the optimization over superdifferentials using a single graph cut and use the exact divide-and-conquer algorithm. We used the min-cut

implementation from [34]. Since the update equations are easily computable, we have also implemented mean-field for the first experiment. For the other two experiments computing the updates requires exhaustive enumeration and is intractable. The results are shown on Figure 1 and the experiments are explained below. We plot the averages of several repetitions of the experiments. Note that computing intervals for marginals requires two MAP-like optimizations per variable; hence we focus on small problems with $|V| = 100$. We point out that obtaining a single factorized approximation (as produced, e.g., by mean-field), only requires *a single MAP-like optimization*, which can be done for more than 270,000 variables [19].

**Log-supermodular: Cuts / Pairwise MRFs.** Our first experiment evaluates L-FIELD on a sequence of distributions that are increasingly more concentrated. Motivated by applications in semi-supervised learning, we sampled data from a 2-dimensional Gaussian mixture model with 2 clusters. The centers were sampled from $\mathcal{N}([3,3], I)$ and $\mathcal{N}([-3,-3], I)$ respectively. For each cluster, we sampled $n = 50$ points from a bivariate normal. These $2n$ points were then used as nodes to create a graph with weight between points $\mathbf{x}$ and $\mathbf{x}'$ equal to $e^{-||\mathbf{x}-\mathbf{x}'||}$. As prior we chose $P(A) \propto \exp(-F(A))$, where $F$ is the cut function in this graph, hence $P(A)$ is a regular MRF. Then, for $k = 1, \ldots, n$ we consider the conditional distribution on the event that $k$ points from the first cluster are on one side of the cut and $k$ points from the other cluster are on the other side. As we provide more evidence, the posterior concentrates, and the intervals for both the log-partition function and marginals shrink. Compared with ground truth, the estimates of the marginal probabilities improve as well. Due to non-convexity, mean-field occasionally gets stuck in local optima, resulting in very poor marginals. To prevent this, we chose the best run out of 20 random restarts. These best runs produced slightly better marginals than L-FIELD for this model, at the cost of less robustness.

**Log-supermodular: Decomposable functions.** Our second experiment assesses the performance as a function of the curvature of $F$. It is motivated by a problem in outbreak detection on networks. Assume that we have a graph $G = (V, E)$ and some of its nodes $E \subseteq V$ have been infected by some contagious process. Instead of $E$, we observe a noisy set $N \subseteq V$, corrupted with a false positive rate of 0.1 and a false negative rate of 0.2. We used a log-supermodular prior $P(A) \propto \exp\left(-\sum_{v \in V} \left(\frac{|N_v \cap A|}{|N_v|}\right)^\mu\right)$, where $\mu \in [0,1]$ and $N_v$ is the union of $v$ and its neighbors. This prior prefers smaller sets and sets that are more clustered on the graph. Note that $\mu$ controls the preference of clustered nodes and affects the curvature. We sampled random graphs with 100 nodes from a Watts-Strogatz model and obtained $E$ by running an independent cascade starting from 2 random nodes. Then, for varying $\mu$, we consider the posterior, which is log-supermodular, as the noise model results in a modular likelihood. As the curvature increases, the intervals for both the log-partition function and marginals decrease as expected. Surprisingly, the marginals are very accurate ($< 0.1$ average error) even for very large curvature. This suggests that our curvature dependent bounds are very conservative, and much better performance can be expected in practice.

**Log-submodular: Facility location modeling.** Our last experiment evaluates how accurate L-FIELD is when quantifying uncertainty in submodular maximization tasks. Concretely, we consider the problem of sensor placement in water distribution networks, which can be modeled as submodular maximization [35]. More specifically, we have a water distribution network and there are some junctions $V$ where we can put sensors that can detect contaminated water. We also have a set $\mathcal{I}$ of contamination scenarios. For each $i \in \mathcal{I}$ and $j \in V$ we have a utility $C_{i,j} \in [0,1]$, that comes from real data [35]. Moreover, as the sensors are expensive, we would like to use as few as possible. We use the facility-location model, more precisely $P(\mathbf{S} = A) \propto \exp(F(A) - 2|A|)$, with $F(A) = \sum_{i \in \mathcal{N}} \max_{j \in A} C_{i,j}$. Instead of optimizing for a fixed placement, here we consider the problem of *sampling* from $P$ in order to quantify the uncertainty in the optimization task. We used the following sampling strategy. We consider nodes $v \in V$ in some order. We then sample a Bernoulli $Z$ with probability $P(Z = 1) = q_v$ based on the factorized distribution $\mathbf{q}$ from the modular upper bound. We then condition on $v \in \mathbf{S}$ if $Z = 1$, or $v \notin \mathbf{S}$ if $Z = 0$. In the computation of the lower bound we used the subgradient $\mathbf{s}^g$ computed from the greedy order of $V$ — the $i$-th element in this order $v_1, \ldots, v_n$ is the one that gives the highest improvement when added to the set formed by the previous $i - 1$ elements. Then, $\mathbf{s}^g \in \partial_F(\emptyset)\colon s_i^g = F(v_i|\{v_0, \ldots, v_{i-1}\})$. We repeated the experiment several times using randomly sampled 500 contamination scenarios and 100 locations from a larger dataset. Note that our approximations get better as we condition on more information (i.e., proceed through the iterations of the sampling procedure above). Also note that even from the very beginning, the marginals are very accurate ($< 0.1$ average error).

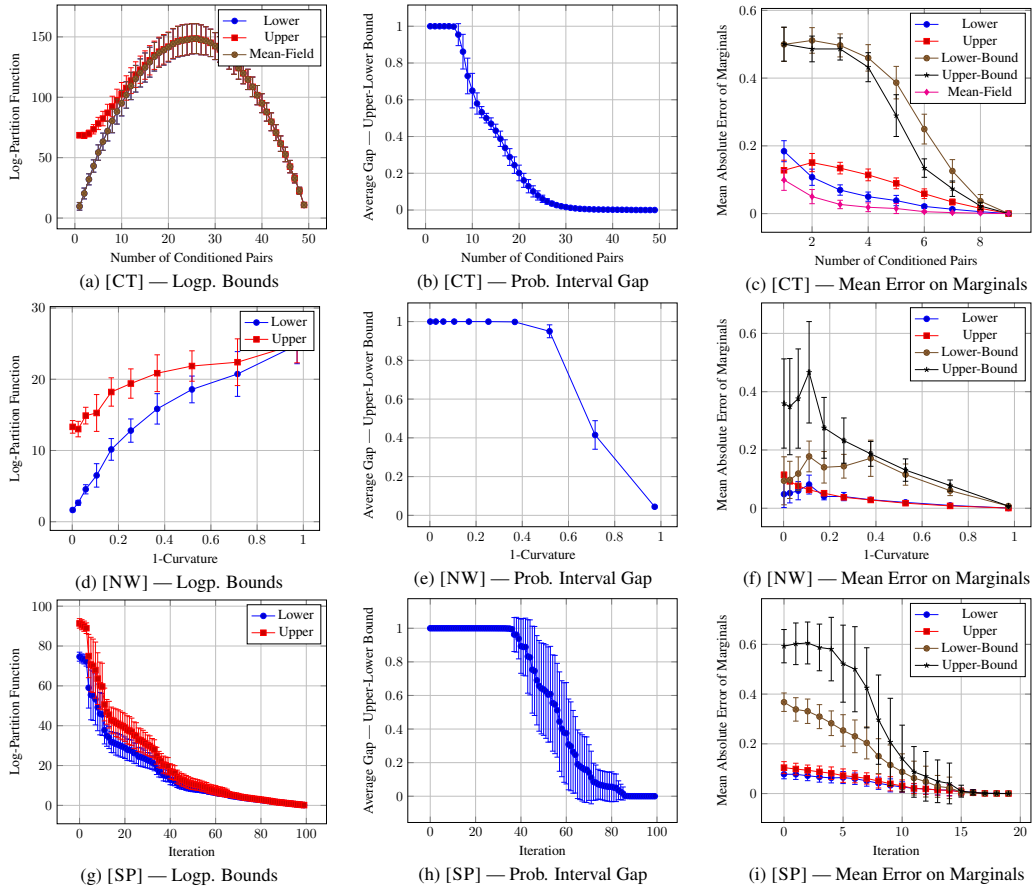

Figure 1: Experiments on [CT] Cuts *(a-c)*, [NW] network detection *(d-f)*, [SP] sensor placement *(g-i)*. Note that to generate *(c,f,i)* we had to compute the exact marginals by exhaustive enumeration. Hence, these three graphs were created using a smaller ground set of size 20. The error bars capture 3 standard errors.

# 8 Conclusion

We proposed L-FIELD, the first variational method for approximate inference in general Bayesian submodular and supermodular models. Our approach has several attractive properties: It produces rigorous upper and lower bounds on the log-partition function and on marginal probabilities. These bounds can be optimized efficiently via convex and submodular optimization. Accurate factorial approximations can be obtained at the same computational cost as performing MAP inference in the underlying model, a problem for which a vast array of scalable methods are available. Furthermore, we identified a natural connection to the traditional mean-field method and bounded the quality of our approximations with the curvature of the function. Our experiments demonstrate the accuracy of our inference scheme on several natural examples of Bayesian submodular models. We believe that our results present a significant step in understanding the role of submodularity – so far mainly considered for optimization – in approximate Bayesian inference. Furthermore, L-FIELD presents a significant advance in our ability to perform probabilistic inference in models with complex, high-order dependencies, which present a major challenge for classical techniques.

**Acknowledgments.** This research was supported in part by SNSF grant 200021_137528, ERC StG 307036 and a Microsoft Research Faculty Fellowship.

## Footnotes

[1] The functions $F(A)$ and $F(A) + c$ encode the same distributions by virtue of normalization.

[2] In the appendix we also consider cardinality constraints, i.e., distributions over sets $A$ that satisfy $|A| \leq k$.

[3]We consider log-supermodular distributions, as the log-submodular case is analogous.

[4]Such submodular loss functions $L$ have been considered, e.g., in document summarization [4].

[5]We differ from the convention to remove $i \in V$ s.t. $G(\{i\}) = 0$. Please see the appendix for a discussion.

[6]The code will be made available at http://las.ethz.ch.

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
