[Supplementary Material]

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

# Appendix

## A  Closure properties for inference in Bayesian submodular models

We first give a brief derivation that shows that the conditional distribution is the one we claimed.

$$P(\mathbf{S} = A | \mathbf{S} \in [X, Y]) = \frac{P(\mathbf{S} \in [X, Y] | S) P(\mathbf{S} = A)}{P(\mathbf{S} \in [X, Y])} = \begin{cases} \exp(-F(A))/\mathcal{Z}_X^Y & \text{if } A \in [X, Y] \\ 0 & \text{otherwise} \end{cases} \tag{10}$$

Similarly to the conditioning that the set is in some lattice, we would also like to point out that using similar computations you also condition on different events. For example, if we condition on the event that the complement of the set is in $[X, Y]$ we have that

$$P(V - \mathbf{S} \in [X, Y]) = \frac{1}{\mathcal{Z}} \sum_{X \subseteq V - A \subseteq Y} \exp(-F(A)) = \frac{1}{\mathcal{Z}} \sum_{X \subseteq B \subseteq Y} \exp(-F(V - B)).$$

The function $G(B) = F(V - B)$ is submodular, and the computation again reduces to computing normalization constants of submodular functions.

## B  Proofs for our variational approach

*Proof of Lemma 1.* Let $g(A) = s(A) + c$ for a modular $s(\cdot)$ be an upper or a lower bound on $F$. Then, using $n = |V|$, we have that

$$\log \mathcal{Z} = \log \sum_{A \subseteq V} \exp(-F(A)) \bigcirc \log \sum_{A \subseteq V} \exp(-s(A) - c) \overset{(a)}{=} -c + \log \sum_{\mathbf{x} \in \{0,1\}^n} \exp(-\mathbf{s}^T \mathbf{x})$$

$$= -c + \log \sum_{\mathbf{x} \in \{0,1\}^n} \prod_{i=1}^{n} \exp(-s_i)^{x_i} = -c + \log \prod_{i=1}^{n} (1 + \exp(-s_i)) = \log \mathcal{Z}^-(\mathbf{s}, c),$$

where $\bigcirc$ is $\leq$ for a lower bound and $\geq$ for an upper bound. In step (a) we have used the correspondence between subsets of $V$ and $\{0, 1\}^n$ vectors. The other case follows analogously.

$$\log \mathcal{Z} = \log \sum_{A \subseteq V} \exp(F(A)) \bigcirc \log \sum_{A \subseteq V} \exp(s(A) + c) = c + \log \sum_{\mathbf{x} \in \{0,1\}^n} \exp(\mathbf{s}^T \mathbf{x})$$

$$= c + \log \sum_{\mathbf{x} \in \{0,1\}^n} \prod_{i=1}^{n} \exp(s_i)^{x_i} = c + \log \prod_{i=1}^{n} (1 + \exp(s_i)) = \log \mathcal{Z}^+(\mathbf{s}, c),$$

where $\bigcirc$ is $\leq$ for an upper bound and $\geq$ for a lower bound. $\qquad\square$

**Approximation to Marginals.** The marginals under a modular distribution can be easily computed as follows. If $F(A) = \frac{1}{\mathcal{Z}} \exp(-s(A) - c)$ for some modular $\mathbf{s}$, then for any $i \in V$

$$P(i \in \mathbf{S}) = \frac{1}{\mathcal{Z}} \sum_{A \subseteq V - \{i\}} e^{-s_i - s(A) - c} = \frac{e^{-s_i - c}}{\mathcal{Z}} \sum_{A \subseteq V - \{i\}} e^{-s(A)}$$

$$= \frac{e^{-s_i - c}}{\mathcal{Z}} \prod_{j \in V - \{i\}} (1 + e^{-s_j}) = \frac{e^{-s_i - c}}{e^{-c} \prod_{j \in V} (1 + e^{-s_j})} \prod_{j \in V - \{i\}} (1 + e^{-s_j})$$

$$= \frac{e^{-s_i}}{1 + e^{-s_i}} = \frac{1}{1 + e^{s_i}}.$$

### B.1  Optimizing over subgradients

Before showing the proofs, we first have to make two definitions. Let $F : 2^V \to \mathbb{R}$ be a normalized submodular function. The *submodular polyhedron* of $F$ is defined as

$$P(F) = \{\mathbf{s} \in \mathbb{R}^V \mid \forall A \subseteq V \colon s(A) \leq F(A)\}. \tag{11}$$

Note that $P(F) = \partial_\emptyset(F)$. For any $X \subseteq V$ we also define the submodular function $\overline{F} : 2^V \to \mathbb{R}$ as $\overline{F}(A) = F(V - A) - F(V)$.

**Theorem 3.** *Let $F : 2^V \to \mathbb{R}$ be a normalized submodular function and let $X \subseteq V$. Then*

$$\partial_F(X) = (-\partial_{\overline{F^X}}(\emptyset)) \times \partial_{F_X}(\emptyset).$$

*Proof.* This follows from several results in the book by Fujishige [8], but for completeness we give a short proof here. From [8][Lem. 6.5] we know that

$$\partial_F(X) = \partial_{F^X}(X) \times \partial_{F_X}(\emptyset).$$

The result follows from the following lemma. $\qquad\square$

**Lemma 5.** *Let $G : 2^U \to \mathbb{R}$ be a normalized submodular function. Then $\partial_G(U) = -\partial_{\overline{G}}(\emptyset)$.*

*Proof.* Let $\mathbf{s} \in \partial_{\overline{G}}(\emptyset)$. This is true iff for all $A \subseteq U$

$$s(A) \leq \overline{G}(A) \iff s(A) \leq G(U - A) - G(U) \iff s(A) + G(U) \leq G(U - A).$$

As this holds for all $A \subseteq U$, for any $B \subseteq U$ we can set $A = U - B$, in which case $U - (U - B) = B$ and we get ($s$ is modular so $s(U - B) = s(U) - s(B)$)

$$s(U - B) + G(U) \leq G(B) \iff -s(B) + s(U) + G(U) \leq G(B).$$

By reordering the terms we get $G(B) + s(B) \geq s(U) + G(U)$. If we compare this with the definition of $\partial_G(U)$ we get that $\partial_{\overline{G}}(\emptyset) = -\partial_G(U)$. $\qquad\square$

**Lemma 6.** *The problem of minimizing $\mathcal{Z}_X^-(\mathbf{s})$ subject to $\mathbf{s} \in \partial_F(X)$ is equivalent to the following strictly convex separable problem*

$$\underset{\mathbf{s}}{\text{minimize}} - F(X) + \sum_{i \in V} \log(1 + e^{-s_i}) \quad \text{subject to} \quad \mathbf{s} \in B(\overline{F^X}) \times B(F_X). \qquad (12)$$

*Proof.* From Theorem 3 we know that

$$\partial_F(X) = (-\partial_{\overline{F^X}}(\emptyset)) \times \partial_{F_X}(\emptyset).$$

Remember that $\mathcal{Z}_X^-(\mathbf{s}) = \mathcal{Z}^-(\mathbf{s}, F(X) - s(X))$, and hence the original problem is equivalent to

$$\underset{\mathbf{s}_X \in -\partial_{\overline{F^X}}(\emptyset)}{\min} \underbrace{-F(X) + \mathbf{s}_X(X)}_{-c} + \sum_{i \in X} \log(1 + e^{-s_i}) + \underset{\mathbf{s}_{V-X} \in \partial_{F_X}(\emptyset)}{\min} \sum_{i \in V-X} \log(1 + e^{-s_i}), \qquad (13)$$

which can be rewritten as

$$-F(X) + \underset{\mathbf{s}_X \in -\partial_{\overline{F^X}}(\emptyset)}{\min} \sum_{i \in X} \log(1 + e^{s_i}) + \underset{\mathbf{s}_{V-X} \in \partial_{F_X}(\emptyset)}{\min} \sum_{i \in V-X} \log(1 + e^{-s_i}). \qquad (14)$$

Note that

$$\underset{\mathbf{z} \in -\partial_{\overline{F^X}}(\emptyset)}{\min} \sum_{i \in X} \log(1 + \exp(z_i)) = \underset{-\mathbf{z} \in \partial_{\overline{F^X}}(\emptyset)}{\min} \sum_{i \in X} \log(1 + \exp(z_i))$$

$$= \underset{\mathbf{s}_X \in \partial_{\overline{F^X}}(\emptyset)}{\min} \sum_{i \in X} \log(1 + \exp(-s_i)) \quad \text{Define } \mathbf{z} \equiv -\mathbf{s}_X.$$

And the original problem is thus equivalent to

$$\text{minimize}_\mathbf{s} - F(X) + \sum_{i=1}^n \log(1 + e^{-s_i}) \quad \text{subject to} \quad \mathbf{s} \in P(\overline{F^X}) \times P(F_X).$$

It remains to show that the optimum is on the product of base polyhedra. Let $\mathbf{s}^*$ be the optimum on the product of the base polyhedra (exists, as base polyhedra are compact). It satisfies

$$\forall \mathbf{y} \in B(\overline{F^X}) \times B(F_X): \nabla h(\mathbf{s}^*)^T \mathbf{y} > \nabla h(\mathbf{s}^*)^T \mathbf{s}^*.$$

As $\nabla h(\mathbf{s}^*) \prec 0$, we have from [9][Prop. 4.2] that the minimizer of the linear objective $\nabla h(\mathbf{s}^*)$ over the full domain is on the product of base polyhedra, which implies that $\mathbf{s}^*$ is a global optimum. $\qquad\square$

**Lemma 7.** *For any normalized submodular function $F : 2^V \to \mathbb{R}$ we have that $B(\overline{F}) = -B(F)$.*

*Proof.* This is well known, proof given for completeness.

$$
\begin{aligned}
\mathbf{s} \in B(F) &\iff F(V) = s(V) \text{ and } \forall A \subseteq V : s(A) \leq F(A) \\
&\iff F(V) = s(V) \text{ and } \forall A \subseteq V : s(V - A) \leq F(V - A) \\
&\iff F(V) = s(V) \text{ and } \forall A \subseteq V : -s(A) \leq F(V - A) - \underbrace{s(V)}_{=F(V)} \\
&\iff -\overline{F}(V) = s(V) \text{ and } \forall A \subseteq V : -s(A) \leq \overline{F}(A) \\
&\iff -\mathbf{s} \in B(\overline{F})
\end{aligned}
$$

$\square$

*Proof of Lemma 2.* Using Lemma 7 and the fact that $\log(1 + e^x) = x + \log(1 + e^{-x})$ we can rewrite the optimization problem in Lemma 6 as

$$
\underset{\mathbf{s}}{\text{minimize}} - F(X) + s_X(X) + \sum_{i \in V} \log(1 + e^{-s_i}) \quad \text{subject to} \quad \mathbf{s} \in B(F^X) \times B(F_X).
$$

Because $\mathbf{s}_X \in B(F^X)$, we have that $s_X(X) = F(X)$ and the claim follows from the fact that $B(F^X) \times B(F_X) \subseteq B(F)$ [8][Lem. 3.1], which is the domain for $X = \emptyset$. $\square$

**Divide-and-Conquer.** To be able to apply the divide and conquer algorithm, we have to be able to minimize $\sum_{i \in V} \log(1 + e^{-s_i})$ over $\mathbf{s} \in \mathbb{R}^V$ such that $\sum_{i \in V} s_i = F(V)$. We will show that the optimal vector has values equal to $F(V)/n$. The Lagrangian is equal to

$$
\mathcal{L}(\mathbf{s}, \lambda) = \sum_{i=1}^{n} \log(1 + e^{-s_i}) + \lambda(\sum_i s_i - F(V)).
$$

Setting the derivative with respect to $s_i$ to zero gives us

$$
\frac{1}{1 + e^{s_i}} = \lambda.
$$

Now, if we set $\lambda^* = 1/(e^{F(v)/n} + 1)$ and $\mathbf{s}^* = (F(V)/n, \ldots, F(V)/n)$, then they satisfy the equation, are primal-dual feasible and by the KKT conditions are also primal-dual optimal, from which the claim follows.

**Derivation of the Dual Problem.** We will now show the dual for the more general problem in Lemma 6. Then, Lemma 3 follows for $X = \emptyset$.

**Lemma 8.** *The Fenchel dual problem of Lemma 6 is*

$$
\underset{\mathbf{p} \in [0,1]^V}{\text{maximize}} \quad \mathbb{H}[\mathbf{p}] - \overline{f^X}(\mathbf{p}_X) - f_X(\mathbf{p}_{V-X}) - F(X), \tag{15}
$$

*where $\overline{f^X}(\mathbf{p}_X)$ and $f_X$ are the Lovász extensions of $\overline{F^X}$ and $F_X$ respectively. Moreover, there is zero duality gap and the pair $(\mathbf{s}^*, \mathbf{p}^*)$ is primal-dual optimal iff*

$$
\mathbf{p}^* = \left(\frac{1}{1 + \exp(s_i^*)}, \ldots, \frac{1}{1 + \exp(s_n^*)}\right) \quad \text{and} \quad \mathbf{p}^{*T}\mathbf{s}^T = \overline{f^X}(\mathbf{p}_X^*) + f_X(\mathbf{p}_{V-X}^*). \tag{16}
$$

*Proof.* The proof is using the same arguments as in [9][§8]. Remember that the referred problem was

$$
\text{minimize}_{\mathbf{s}} - F(X) + \sum_{i \in V} \log(1 + e^{-s_i}) \quad \text{subject to} \quad \mathbf{s} \in B(\overline{F^X}) \times B(F_X).
$$

Let us denote by $I_C(\mathbf{s})$ the indicator function on the set $C$, equal to 0 on $C$ and $+\infty$ elsewhere. We also define $\phi(x) = \log(1 + e^{-x})$. We can then rewrite the problem as minimizing the following function over $\mathbf{s} \in \mathbb{R}^V$

$$
-F(X) + \sum_{i \in X} \phi(s_i) + \sum_{j \notin X} \phi(s_j) + I_{B(\overline{F^X})}(\mathbf{s}_X) + I_{B(F_X)}(\mathbf{s}_{V-X}).
$$

Note that the conjugate of the indicator function over the base polyhedron is the Lovász extension $I_{B(F)}^*(\mathbf{p}) = \sup_{\mathbf{s} \in B(F)} \mathbf{s}^T \mathbf{p} = f(\mathbf{p})$. The function $\phi$ has the following conjugate

$$\phi^*(p) = \begin{cases} \underbrace{-p\log(-p) + (1+p)\log(1+p)}_{-\mathbb{H}[-p]} & \text{if } p \in [-1,0] \\ \infty & \text{otherwise} \end{cases},$$

where $\mathbb{H}[p]$ is the entropy of a Bernoulli random variable with success probability $p$. Define

$$h(\mathbf{s}) = I_{B(\overline{F^X})}(\mathbf{s}_X) + I_{B(F_X)}(\mathbf{s}_{V-X}) \text{ and } g(\mathbf{s}) = \sum_{i \in V} \phi(s_i).$$

The Fenchel dual problem is equal to

$$\text{maximize}_{\mathbf{p}} \; -g^*(\mathbf{p}) - h^*(-\mathbf{p}).$$

Note that $h$ and $g$ are sums of functions which depend on different variables. Hence, they are sums of independent functions and their conjugates are equal to the sum of the conjugates of the summands [36][§3.3.2], i.e. the problem becomes

$$\text{maximize}_{\mathbf{p} \in [-1,0]^n} \underbrace{\sum_{i=1}^n \mathbb{H}[-p_i]}_{-g^*(\mathbf{p})} - (\underbrace{\overline{f^X}(\mathbf{p}_X) + f_X(\mathbf{p}_{V-X})}_{h^*(-\mathbf{p})}).$$

If we denote by $\mathbb{H}[\mathbf{p}] = \sum_{i=1}^n \mathbb{H}[p_i]$ to be entropy of a Bernoulli random vector with element probabilities $\mathbf{p}$ and we re-parametrize the problem in $-\mathbf{p}$ we get the following equivalent form

$$\text{maximize}_{\mathbf{p} \in [0,1]^n} \; \mathbb{H}[\mathbf{p}] - \overline{f^X}(\mathbf{p}) - f_X(\mathbf{p}).$$

The function $g$ is smooth on its domain $\mathbb{R}^n$ and this is sufficient to have no duality gap [37]. Moreover, from Theorem 2 in the same paper we know that $(\mathbf{s}^*, \mathbf{p}^*)$ is primal-dual optimal iff $\mathbf{p}^* \in (-\partial g(\mathbf{s}^*)) \cap \partial h(\mathbf{s}^*) \overset{g \text{ diff.}}{=} \{-\nabla g(\mathbf{s}^*)\} \cap \partial h(\mathbf{s}^*)$. Note that this means that $\mathbf{p}^* = -\nabla g(\mathbf{s}^*)$ and this is exactly the first part of the claim. To show the second part, remember that for the indicator function $I_C(\cdot)$ the subdifferential at point $\mathbf{x}$ is the set of all linear functions that achieve their maximum over $C$ at $\mathbf{x}$ [37]. Hence, as $h$ is the indicator on the set, we have that $B(\overline{F^X}) \times B(F_X)$

$$\partial h(\mathbf{s}^*) = \{\mathbf{p} \colon \mathbf{p}^T \mathbf{s}^* = \sup_{\mathbf{s} \in B(\overline{F^X}) \times B(F_X)} \mathbf{p}^T \mathbf{s}\}$$

$$= \{\mathbf{p} \colon \mathbf{p}^T \mathbf{s}^* = \underbrace{\sup_{\mathbf{s}_X \in B(\overline{F^X})} \mathbf{p}_X^T \mathbf{s}_X}_{\overline{f^X}(\mathbf{p}_X)} + \underbrace{\sup_{\mathbf{s}_{V-X} \in B(F_X)} \mathbf{p}_{V-X}^T \mathbf{s}_{V-X}}_{f_X(\mathbf{p}_{V-X})}\}.$$

$\square$

## B.2 Optimizing over supergradients

*Proof of Lemma 4.* Substituting the bar-supergradient $\overline{\mathbf{s}}^X \in \partial^F(X)$ in the definitions for $\mathcal{Z}_X^+$ and $\mathcal{Z}_X^-$ gives us the following equations.

$$\log Z_X^+(\overline{\mathbf{s}}^X) = \underbrace{F(X) - \sum_{i \in X} F(i|V-i)}_{\overline{\mathbf{s}}^X(X)} + \sum_{i \in X} \log(1 + \exp(F(i|V-i))) + \sum_{i \notin X} \log(1 + \exp(F(i)))$$

$$\log Z_X^-(\overline{\mathbf{s}}^X) = \underbrace{-F(X) + \sum_{i \in X} F(i|V-i)}_{\overline{\mathbf{s}}^X(X)} + \sum_{i \in X} \log(1 + \exp(-F(i|V-i))) + \sum_{i \notin X} \log(1 + \exp(-F(i)))$$

We can simplify the equations to the the following ones.

$$\log Z_X^+(\overline{\mathbf{s}}) = F(X) + \sum_{i \in X} \log(1 + \exp(-F(i|V-i))) + \sum_{i \notin X} \log(1 + \exp(F(i)))$$

$$\log Z_X^-(\overline{\mathbf{s}}) = -F(X) + \sum_{i \in X} \log(1 + \exp(F(i|V-i))) + \sum_{i \notin X} \log(1 + \exp(-F(i)))$$

Note that any $m : 2^V \to \mathbb{R}$ given as $m(A) = \sum_{i \in A} a_i + \sum_{j \notin A} b_j$ for some numbers $a_i, b_j$ is a modular plus constant, because this is equivalent to $\sum_{i \in A} a_i + \sum_{i \in V} b_i + \sum_{i \in A}(-b_i)$. The claims follow by writing the modular term in the standard form and ignoring the constant $\sum_{i \in V} b_i$. $\qquad \square$

## C  Mean-field methods and the multi-linear extension

For completeness we will derive the mean-field update equation. It is well known that the partial derivatives of the multi-linear extension [31] are constant functions and equal to

$$\partial_{q_i} \mathbb{E}_{\mathbf{q}}[F(A)] = \mathbb{E}_{\mathbf{q}_{+i}}[F(A)] - \mathbb{E}_{\mathbf{q}_{-i}}[F(A)],$$

where $\mathbf{q}_{+i}$ is a distribution over $V$ such that $j \neq i$ is selected with probability $q_j$ and the element $i$ is picked with probability 1, and $q_{-i}$ is analogously defined over $V - \{i\}$ (i.e. $i$ is never sampled). Because the entropy is concave and the multi-linear extension is linear in each argument, we can perform coordinate ascent by picking $q_i$ so that the partial derivative vanishes, i.e.

$$- \underbrace{(\mathbb{E}_{\mathbf{q}_{+i}}[F(A)] - \mathbb{E}_{\mathbf{q}_{-i}}[F(A)])}_{\Delta_i} + \log(1 - q_i) - \log q_i = 0.$$

Which means that we have to set $q_i'$ to

$$q_i' = \frac{e^{-\Delta_i}}{e^{-\Delta_i} + 1} = \frac{1}{1 + e^{\Delta_i}}.$$

And this is the update equation for the naïve mean field method. Let us analyze $\Delta_i$, which is equal to

$$\Delta_i = \sum_{A \subseteq V - i} P_{\mathbf{q}-i}(A) F(A \cup \{i\}) - \sum_{A \subseteq V - i} P_{\mathbf{q}-i}(A) F(A)$$

$$= \sum_{A \subseteq V - i} P_{\mathbf{q}-i}(A) F(i|A) = \mathbb{E}_{A \sim \mathbf{q}-i}[F(i|A)].$$

And we get an interpretation in terms of the marginal gains. Note that computing these updates exactly is intractable for general functions $F$.

## D  Proofs for the curvature-dependent approximation bounds

We would first like to point out a small technical difference. The usually made assumption on the polymatroid function is that $\forall i \in V : G(\{i\}) > 0$ — because, as observed by [38], $i$ has no effect on the function values and can be safely removed from the ground set. Because we do not work with a submodular optimization problem, we do not remove such elements. However, as we will only use the modular bounds on $G$, we can easily lift the results from the restriction of $G$ to $\{i \in V \mid G(\{i\}) > 0\}$.

**Lemma 9.** *Let $F(A) = G(A) + m(A)$ be submodular, where $G(\cdot)$ is polymatroid and $m(\cdot)$ is modular. Let $s(A) + c$ for some modular function $s(\cdot)$ be a lower bound on $G$ and assume that for some $\eta \in [0, 1]$ we have $\forall A \subseteq V, (1 - \eta)G(A) \leq s(A) + c \leq G(A)$. Then*

$$\log \mathcal{Z}^-(\mathbf{m} + \mathbf{s}, c) - \log \sum_{A \subseteq V} \exp(-F(A)) \leq \eta G_{\text{MAX}}, \text{ and} \tag{17}$$

$$\log \sum_{A \subseteq V} \exp(+F(A)) - \log \mathcal{Z}^+(\mathbf{m} + \mathbf{s}, c) \leq \eta G_{\text{MAX}}. \tag{18}$$

*Proof.*

$$\log \mathcal{Z}^-(\mathbf{m}+\mathbf{s},c) - \log \sum_{A\subseteq V}\exp(-F(A)) = \log \frac{\sum_{A\subseteq V}\exp(-(m(A)+s(A)+c))}{\sum_{A\subseteq V}\exp(-F(A))}$$

$$\leq \log \frac{\sum_{A\subseteq V}\exp(-(m(A)+(1-\eta)G(A)))}{\sum_{A\subseteq V}\exp(-F(A))} \quad \text{By assumption.}$$

$$= \log \frac{\sum_{A\subseteq V}\exp(-F(A)+\eta G(A))}{\sum_{A\subseteq V}\exp(-F(A))}$$

$$\leq \log \frac{\sum_{A\subseteq V}\exp(-F(A)+\eta G_{\text{MAX}})}{\sum_{A\subseteq V}\exp(-F(A))} \quad \eta \geq 0,\ F(A)\leq G_{\text{MAX}}.$$

$$= \eta G_{\text{MAX}}$$

The other part is analogously proven.

$$\log \sum_{A\subseteq V}\exp(+F(A)) - \log \mathcal{Z}^+(\mathbf{m}+\mathbf{s},c) = \log \frac{\sum_{A\subseteq V}\exp(F(A))}{\sum_{A\subseteq V}\exp(m(A)+s(A)+c)}$$

$$\leq \log \frac{\sum_{A\subseteq V}\exp(F(A))}{\sum_{A\subseteq V}\exp(m(A)+(1-\eta)G(A))} \quad \text{By assumption.}$$

$$= \log \frac{\sum_{A\subseteq V}\exp(F(A))}{\sum_{A\subseteq V}\exp(F(A))\exp(-\eta G(A))}$$

$$\leq \eta G_{\text{MAX}} \quad \eta \geq 0,\ F(A)\leq G_{\text{MAX}}.$$

$\square$

*Proof of Theorem 1.* The claim that $\mathbf{s}+\mathbf{m}$ is in $\partial_F(\emptyset)$ follows easily. Because $\mathbf{s}\in\partial_G(\emptyset)$, we have that $\forall A \subseteq V\colon s(A) \leq G(A) \implies \forall A \subseteq V\colon s(A)+m(A) \leq G(A)+m(A) = F(A)$. We will complete the proof by showing that the criterion in the previous lemma is satisfied for the given subgradient with $\eta = \kappa$, which implies the claim. Note that we have

$$\forall j \in V\colon G(j|V-\{j\}) \geq (1-\kappa)G(\{j\}), \tag{19}$$

where the claim follows from the definition of curvature if $G(\{j\}) > 0$. Otherwise, because $G$ is polymatroid, we have that $0 \leq G(j|V-\{j\}) = G(\{j\}) = 0$. Then, for any $A \subseteq V$

$$s(A) = \sum_{i\in A}[G(S_i^\sigma) - G(S_{i-1}^\sigma)]$$

$$= \sum_{i\in A}G(i|S_{i-1}^\sigma) \quad \text{Definition of marginal gain.}$$

$$\geq \sum_{i\in A}G(i|V-\{i\}) \quad \text{Submodularity of } G \text{ and } S_{i-1}^\sigma \subseteq V-\{i\}.$$

$$\geq (1-\kappa)\sum_{i\in A}G(\{i\}) \quad \text{From eq. (19).}$$

$$\geq (1-\kappa)G(A). \quad z(A)=\sum_{i\in A}G(\{i\}) \text{ is in } \partial^G(\emptyset).$$

$\square$

**Lemma 10.** *Let $F(A) = G(A) + m(A)$ be submodular, where $G(\cdot)$ is polymatroid and $m(\cdot)$ is modular. Let $s(A) + c$ for some modular function $s$ be an upper bound on $G$ and assume that for some $\eta \geq 0$ we have $\forall A \subseteq V\colon G(A) \leq s(A)+c \leq (1+\eta)G(A)$. Then*

$$\log \sum_{A\subseteq V}\exp(-F(A)) - \log \mathcal{Z}^-(\mathbf{m}+\mathbf{s},c) \leq \eta G_{\text{MAX}},\ and \tag{20}$$

$$\log \mathcal{Z}^+(\mathbf{m}+\mathbf{s},c) - \log \sum_{A\subseteq V}\exp(+F(A)) \leq \eta G_{\text{MAX}}. \tag{21}$$

*Proof.* We will use the following bound that follows from the assumption $s(A)+c \leq G(A)+\eta G_{\text{MAX}}$.

$$\log \sum_{A \subseteq V} \exp(-F(A)) - \log \mathcal{Z}^-(\mathbf{m}+\mathbf{s},c) = \log \frac{\sum_{A \subseteq V} \exp(-F(A))}{\sum_{A \subseteq V} \exp(-(m(A)+s(A)+c))}$$

$$\leq \log \frac{\sum_{A \subseteq V} \exp(-F(A))}{\sum_{A \subseteq V} \exp(-(m(A)+G(A)+\eta G_{\text{MAX}}))} \qquad \text{Bound above.}$$

$$= \log \frac{\sum_{A \subseteq V} \exp(-F(A))}{\sum_{A \subseteq V} \exp(-F(A))\exp(-\eta G_{\text{MAX}})}$$

$$= \eta G_{\text{MAX}}$$

The other part is analogously proven.

$$\log \mathcal{Z}^+(\mathbf{s},c) - \log \sum_{A \subseteq V} \exp(F(A)) = \log \frac{\sum_{A \subseteq V} \exp(m(A)+s(A)+c)}{\sum_{A \subseteq V} \exp(F(A))}$$

$$\leq \log \frac{\sum_{A \subseteq V} \exp(m(A)+G(A)+\eta G_{\text{MAX}})}{\sum_{A \subseteq V} \exp(F(A)} \qquad \text{Bound above.}$$

$$= \log \frac{\sum_{A \subseteq V} \exp(\eta G_{\text{MAX}})\exp(F(A))}{\sum_{A \subseteq V} \exp(F(A))}$$

$$\leq \eta G_{\text{MAX}} \qquad\qquad \eta \geq 0, F(A) \leq G_{\text{MAX}}.$$

$\square$

*Proof of Theorem 2.* Let us first show that $\mathbf{s} + \mathbf{m} \in \partial^F(\emptyset)$. This is easily seen to hold because $\mathbf{s} \in \partial^G(\emptyset)$, which implies that $\forall A \subseteq V \colon s(A) \geq G(A) \implies \forall A \subseteq V \colon s(A) + m(A) \geq G(A) + m(A) = F(A)$.

The first inequality in the theorem follows from the previous lemma and [38][Lem. 3.1], where it is shown that the chosen $s$ satisfies the above condition with $\eta = \kappa/(1-\kappa)$. Note that in the cited paper they consider only polymatroid functions that are non-zero on singleton elements. We can apply their result to $G(\cdot)$ and $s(\cdot)$ restricted on the set $\overline{V} = \{i \in V \mid G(\{i\}) > 0\}$. Then, for any $A \subseteq V$, $G(A) = G(\overline{V} \cap A)$, and similarly, $s(A) = \sum_{i \in A} G(\{i\}) = \sum_{i \in \overline{V} \cap A} G(\{i\}) = s(\overline{V} \cap A)$, from which the claim follows. $\square$

## E  Experiments

Here we provide more information on the experiments.

**Decomposable Functions.** The data was generated using a Watts-Strogatz model with 100 nodes and parameters: $k = 4$ (number of initial neighbors), and rewiring probability $\rho = 0.7$. The independent cascade model had the probability parameter set to $0.1$. Let us also precisely state the likelihood model. Let $N$ be the observed noisy set. If we set $\alpha$ to be the false negative rate (active $\to$ non-active) and $\beta$ is the false positive rate (non-active $\to$ active), then for any $i \in V$

$$P(i \in N|A) = \begin{cases} 1-\alpha & \text{if } i \in A \\ \beta & \text{if } i \notin A \end{cases}. \tag{22}$$

Or, if we write it in a different form, we have the following.

$$P(i \in N|A) = (1-\alpha)^{[i \in A]}\beta^{1-[i \in A]} = e^{[i \in A] \log \frac{1-\alpha}{\beta} + \log \beta} \tag{23}$$

$$P(i \notin N|A) = \alpha^{[i \in A]}(1-\beta)^{1-[i \in A]} = e^{[i \in A] \log \frac{\alpha}{1-\beta} + \log(1-\beta)} \tag{24}$$

The modular likelihood is then (the minus in front is because we use a log-supermodular distribution)

$$s(\{i\}) = \begin{cases} -\log \frac{1-\alpha}{\beta} & \text{if } i \in N \\ -\log \frac{\alpha}{1-\beta} & \text{if } i \notin N \end{cases}. \tag{25}$$

## F Cardinality constraints

We will w.l.o.g. assume that $V = \{1, 2, \ldots, n\}$. Let us first show how to evaluate the bounds for log-supermodular functions as the ones for log-submodular easily follow. Let us consider the form of the bounds

$$\mathcal{Z}^-(\mathbf{s}, c) = \exp(-c) \sum_{\mathbf{x} \in \{0,1\}^n} \prod_{i=1}^n \exp(-s_i)^{x_i}.$$

Under a cardinality constraint if we consider sets of size at most $k$, we get the following bound instead.

$$\mathcal{Z}_k^-(\mathbf{s}, c) = \exp(-c) \sum_{\substack{\mathbf{x} \in \{0,1\}^n \\ \|\mathbf{x}\|_1 \leq k}} \prod_{i=1}^n \exp(-s_i)^{x_i}.$$

The $m$-th elementary symmetric polynomial is defined as

$$a_0(x_1, \ldots, x_n) = 1, \qquad a_m(x_1, \ldots, x_n) = \sum_{1 \leq i_1 < i_2 < \ldots < i_m \leq m} x_{i_1} \ldots x_{i_m}$$

By taking the logarithm we can rewrite the bound as

$$\log \mathcal{Z}_k(\mathbf{s}, c) = -c + \log \sum_{m=0}^k a_m(e^{-s_1}, \ldots, e^{-s_n}).$$

Define $p_k(x_1, \ldots, x_n) = \sum_{i=1}^n x_i^k$, then using Newton's identities

$$a_k(x_1, \ldots, x_n) = \frac{1}{k} \sum_{i=1}^k (-1)^{i-1} a_{k-i}(x_1, \ldots, x_n) p_i(x_1, \ldots, x_n).$$

And we can evaluate the bound by recursively computing the elementary symmetric polynomials.

**Frank-Wolfe.** We will show that the analogue of Lemma 2 is again a convex, but non-separable, problem and can be solved by applying the Frank-Wolfe algorithm. Moreover, we will also show that we will obtain the best bound by only considering the subdifferential at $\emptyset$. We define for $\mathbf{s} \in \partial_F(X)$ the shorthand $\mathcal{Z}_{k,X}^-(\mathbf{s}) = \mathcal{Z}_k^-(\mathbf{s}, F(X) - s(x))$.

**Lemma 11.** *For all $X \subseteq V$ we have that $\min_{\mathbf{s} \in \partial F(\emptyset)} \mathcal{Z}_{k,\emptyset}^-(\mathbf{s}) \leq \min_{\mathbf{s} \in \partial F(X)} \mathcal{Z}_{k,X}^-(\mathbf{s})$. The latter problem is equivalent to*

$$\log \sum_{m=0}^k a_m(e^{-s_1}, \ldots, e^{-s_n}) \quad \text{subject to} \quad \mathbf{s} \in B(F). \tag{26}$$

*Proof.* Follows from the following lemma and the fact that $B(F^X) \times B(F_X) \subseteq B(F)$ [8][Lem. 3.1], which is the domain for $X = \emptyset$. $\square$

**Lemma 12.** *For all $X \subseteq V$ we have that $\min_{\mathbf{s} \in \partial_F(X)} \mathcal{Z}_{k,X}^-(\mathbf{s})$ is equivalent to*

$$\text{minimize}_{\mathbf{s}} \log \sum_{m=0}^k a_m(e^{-s_1}, \ldots, e^{-s_n}) \quad \text{subject to} \quad \mathbf{s} \in B(F^X) \times B(F_X). \tag{27}$$

*Proof.* From Theorem 3 we know that the problem $\min_{\mathbf{s} \in \partial_F(X)} \mathcal{Z}_{k,X}^-(\mathbf{s})$ is equivalent to

$$\text{minimize}_{\mathbf{s}} - F(X) + s(X) + \log \sum_{m=0}^k a_m(e^{-s_1}, \ldots, e^{-s_n}) \quad \text{subject to} \quad \mathbf{s} \in (-P(\overline{F^X})) \times P(F_X). \tag{28}$$

Recall that $P(\cdot)$ denotes the submodular polyhedron (see Appendix B.1). Note that each term $a_m(e^{-s_1}, \ldots, e^{-s_n})$ is log-convex. Hence, the objective is convex, as log-convex functions are closed under summation [36][§3.5.2]. Let us analyze the gradient of $\mathcal{Z}_{k,X}^-$.

$$\nabla \log \mathcal{Z}_{k,X}^-(\mathbf{s}) = \begin{pmatrix} \mathbf{1}_X \\ \mathbf{0}_{V-X} \end{pmatrix} - \frac{\sum_{m=0}^k \nabla a_m(e^{-s_1}, \ldots, e^{-s_n})}{\sum_{m=0}^k a_m(e^{-s_1}, \ldots, e^{-s_n})}.$$

Note that $[\nabla a_m(e^{-s_1}, \ldots, e^{-s_n})]_{s_i} = -e^{-s_i} a_{m-1}(e^{-s_1}, \ldots, e^{s_{i-1}}, e^{s_{i+1}}, \ldots, e^{-s_n})$, which implies that we can write the gradient as follows.

$$\nabla \log \mathcal{Z}_{k,X}^-(\mathbf{s}) = \begin{pmatrix} \mathbf{1}_X \\ \mathbf{0}_{V-X} \end{pmatrix} - \frac{\sum_{m=0}^k \begin{pmatrix} e^{-s_1} a_{m-1}(e^{-s_2}, \ldots, e^{-s_n}) \\ \vdots \\ e^{-s_i} a_{m-1}(e^{-s_1}, \ldots, e^{s_{i-1}}, e^{s_{i+1}}, \ldots, e^{-s_n}) \\ \vdots \\ e^{-s_n} a_{m-1}(e^{-s_1}, \ldots, e^{-s_{n-1}}) \end{pmatrix}}{\sum_{m=0}^k a_m(e^{-s_1}, \ldots, e^{-s_n})}$$

First observe that the partial derivatives with respect to $s_i$ for $i \notin X$ are immediately seen to be strictly negative. Otherwise, if $i \in X$, the partial derivative is equal to

$$1 - \frac{\sum_{m=0}^{k-1} e^{-s_i} a_m(e^{-s_1}, \ldots, e^{-s_{i-1}}, e^{-s_{i+1}}, \ldots, e^{-s_n})}{\sum_{m=0}^k a_m(e^{-s_1}, \ldots, e^{-s_n})}.$$

And this is clearly strictly positive, as the numerator denominator contains all summands in the numerator and strictly more. We will now show that the optimum over $\mathcal{B} = (-B(\overline{F^X})) \times B(F_X)$ is a global optimum. Let $(-\mathbf{s}_X^*, \mathbf{s}_{V-X}^*)$ be optimal over $\mathcal{B}$ and let $(-\mathbf{g}_X, \mathbf{g}_{V-X})$ be the gradient at that point, where $\mathbf{g}_X, \mathbf{g}_{V-X} \prec 0$. We have for any $\mathbf{m} \in \mathcal{B}$

$$(-\mathbf{g}_X^*, \mathbf{g}_{V-X}^*)((\mathbf{m}_X, \mathbf{m}_{V-X})^T - (-\mathbf{s}_X^*, \mathbf{s}_{V-X}^*)^T) \geq 0.$$

Hence, $\mathbf{s}_{V-X}^*$ is a minimizer of the linear function $\mathbf{g}_{V-X}^*$ over $B(F_X)$. Because $\mathbf{g}_{V-X}^* \prec 0$, by the same argument as in the proof of Lemma 6, we get that $\mathbf{s}_{V-X}^*$ is optimal over $P(F_X)$. Similarly, $-\mathbf{s}_X^*$ is a minimizer of the linear function $-\mathbf{g}_X$ over $-B(\overline{F^X})$. Or equivalently, $\mathbf{s}_X^*$ is a minimizer of the linear function $\mathbf{g}_X$ over $B(\overline{F^X})$, and the claim follows. Finally, from Lemma 7 we know that $-B(\overline{F^X}) = B(F^X)$, which implies that $s(X) = s_X(X) = F(X)$, and this completes the proof. $\square$