[Reviews · NeurIPS 2014]

Submitted by Assigned_Reviewer_15

The authors present a variational approach L-FIELD to general log-submodular and supermodular distributions. Theoretical contributions include deriving upper and lower bounds on the log-partition function and fully factorized approximate posteriors. The quality of the approximation is tested with respect to the curvature of the function. Empirical results are presented on GMM cuts and MRFs, decomposable functions and facility location modeling.

The paper is well written, has theoretical and empirical contributions.
Summary: The authors present a variational approach for approximate inference in general Bayesian sub and super modular models. The paper is very well written and has significant theoretical and empirical contributions.

Submitted by Assigned_Reviewer_26

This paper investigates an interesting idea of approximating the normalization constant (thereby computing various inference quantities like marginals, conditional distributions etc.) for the general class of log-submodular and log-supermodular distributions.

On the positive side, I thought the idea of this paper was neat -- to use the semigradients to provide upper/lower bounds of the functions, and compute the constants for these bounds. In this way, they leveraged a lot of the existing results concerning approximation bounds, to measure the tightness of the approximation.

A main drawback of this paper is that the results seem theoretically pretty weak, given that the main contribution of this paper is theoretical. Theorem 2 is, in my opinion, the only result which talks about the tightness of the approximation, and provides a factor, which is in the worst case O(n). I wonder if there is a notion of hardness factor in computing the normalization factor (using, for instance, the construction of [Balcan and Harvey, 2009]). In my opinion, the paper is somewhat incomplete without a discussion on this, or even possible directions of computing the normalization factor. Without a notion of hardness, the worst case factor can look very discouraging, and there is no sense of how hard this problem can be (apart from the #P-hardness result, which is stated in lines 150 -- it would be nice to have a reference or a proof of this). Can one say something like, no polynomial time algorithm can compute Z up to a factor of p(n), where p(n) is a polynomial in n?

I would also have loved to see some more discussion on the utility of computing these quantities. For instance, does this give method to approximately sample from a log-submodular distribution? Does this give guarantees in terms of learning mixtures of submodular functions? It seems to me that it would not be hard to show guarantees for Maximum-Likelihood learning of mixtures of submodular functions under this model. Consider the following setup. We have a mixture of submodular functions, and define p(X) \propto \exp(\sum_i w_i f_i(X)). One could then do approximate max-Likelihood learning to learn the weights w. This might be an interesting application of the framework. It would also be nice to see some stronger empirical utility of these models on real world problems, like for e.g. summarization. Does this framework admit sampling algorithms from submodular functions?

On a whole, the paper misses some important aspects of this story. I would really love to see more discussion on this.
Summary: To summarize, I think this paper considers an important problem, and provides a first interesting formulation of it, but stands short in providing a complete picture of this problem.

# Edits after author feedback:
Dear Authors,

My main concerns are that this paper does not as of yet, provide a complete story about log-submodular and log-supermodular distributions. I feel that the results are theoretically weak, and are not adequately supported by empirical evidence. In the rebuttal, the authors point out that the approximate distributions provide a means to sample from these distributions. However, given that these approximations are rather weak, I would not expect them to provide effective provedures for sampling. On a whole, I feel that the authors should empirically demonstrate the effectiveness of this approach via a real world application.

Submitted by Assigned_Reviewer_35

Summary: This paper draws links between submodular functions and
variational inference in Bayesian models. By adopting variational
approximations with a certain modular form, efficient optimization
methods become available, with tight lower and upper bounds that
quantify approximation accuracy. Some nice theoretical work is
presented, along with several experiments that provide some
quantification of the performance of the method.

This is a strong paper. While somewhat dense, I felt that the
material was interesting, well-motivated, theoretically substantive
and practically relevant. I don't think I could ask for more in a
paper.
Summary: An excellent paper with a strong mix of theory and empirical results.
Author Feedback
Author rebuttal: We would like to thank all reviewers for their comments.

Regarding the issues raised by Reviewer #26:

* We wish to emphasize that the contributions of the paper are not only theoretical, but have several important immediate practical applications, as appreciated by the other two reviewers. We treat variable dependencies of *arbitrarily high order*, state variational convex problems that do not suffer from local minima, and give practical algorithms for solving them. For example, when we use super-gradients, we solve the problem at the cost of a single submodular minimization problem. Our methods further provide useful rigorous upper and lower bounds on marginal probabilities. Some of the problem classes that we consider have been extensively used in many domains, as we give some examples in Section 3.
* We do provide means to sample from both log-supermodular and log-submodular distributions (in fact, this is the scenario used for the “facility location modeling” experiment, starting at line #360). Namely, with each of the approximation procedures we suggest, one also obtains a completely factorized approximate distribution that can be easily used for sampling. Let X_i\in {0,1} be the random variable denoting if element i has been chosen in the set. Using the chain rule we have P(X_1,...,X_n)=P(X_1)P(X_2|X_1)...P(X_n|X_1,...X_{n-1}). It is now easy to see how to sample, as each distribution P(X_k|X_1=x_1,..X_{k-1}=x_{k-1}) is again in the same family (as argued on lines #140-142), and we can use our methods to obtain a factorized approximate distribution for it.
* For every distribution in the suggested families, and for any data set, we obtain both lower- and upper-bounds on the marginal log-likelihood, which is the intractable quantity necessary for learning and model selection. The guarantees from Theorems 1 and 2 do translate here, as the the log-partition function appears only additively in the log-likelihood. We do agree that our methods might be useful for the special case of learning mixtures. For example, to compute the gradient with respect to w_i we need to compute the expectation of F_i under the mixture distribution. One way of doing this would be to approximate the expectation by sampling from the factorized distribution. However, given the length restriction, we have left this for future work.
* In fact there are hardness of approximation results for binary pairwise Markov random fields, which are a strict subset of what we consider — for example, in [0, Thm. 1.1] it is shown that in the log-supermodular (ferromagnetic) case, the problem of approximating the log-partition function is of the same hardness (interreducible) as that of counting independent sets in bipartite graphs, which has no known approximation algorithms. The log-submodular case (antiferromagnetic) case is known to have no FPRAS unless RP=NP [1, Thm. 14]. The #P-completeness is also proved in [1, Thm. 15]. We will add a brief discussion and include these references in the final version. Note that as far as we know, none of the existing variational inference algorithms provide *any* approximation guarantees and may suffer from non-convexity and computational intractability (in case of high-order dependencies) as argued in the paper. Hence we believe that our curvature-based results (even if not the main emphasis of the paper) provide substantial progress over the state of the art.

[0] Goldberg, Leslie Ann and Jerrum, Mark. (2007) The complexity of ferromagnetic Ising with local fields. Combinatorics, Probability and Computing.
[1] Mark Jerrum and Alistair Sinclair, (1993) Polynomial-Time Approximation Algorithms for the Ising Model, SIAM J. Comput.